# The Effects of Sulglycotide on the Adhesion and the Inflammation of *Helicobacter Pylori*

**DOI:** 10.3390/ijerph17082918

**Published:** 2020-04-23

**Authors:** Ji Yeong Yang, Pumsoo Kim, Seok-Hoo Jeong, Seong Woong Lee, Yu Sik Myung, Myong Ki Baeg, Jong-Bae Kim

**Affiliations:** 1Department of Biomedical Laboratory Science, College of Health Sciences, Yonsei University, Wonju 26493, Korea; clever1088@nate.com; 2Division of Gastroenterology, Department of Internal Medicine, Catholic Kwandong University International St. Mary’s Hospital, Incheon 22711, Korea; pskimgi@naver.com (P.K.); dittomyung@gmail.com (Y.S.M.); baegmk@gmail.com (M.K.B.); 3Samil Pharmaceutical Co. Ltd., Seoul 06666, Korea; citos@samil-pharm.com

**Keywords:** adhesion, *helicobacter pylori*, inflammation, sulglycotide

## Abstract

*Helicobacter pylori* (*H. pylori*) is a primary etiologic factor in gastric diseases. Sulglycotide is a glycopeptide derived from pig duodenal mucin. Esterification of its carbohydrate chains with sulfate groups creates a potent gastroprotective agent used to treat various gastric diseases. We investigated the inhibitory effects of sulglycotide on adhesion and inflammation after *H. pylori* infection in human gastric adenocarcinoma cells (AGS cells). *H. pylori* reference strain 60190 (ATCC 49503) was cultured on Brucella agar supplemented with 10% bovine serum. Sulgylcotide-mediated growth inhibition of *H. pylori* was evaluated using the broth dilution method. Inhibition of *H. pylori* adhesion to AGS cells by sulglycotide was assessed using a urease assay. Effects of sulglycotide on the translocation of virulence factors was measured using western blot to detect cytotoxin-associated protein A (CagA) and vacuolating cytotoxin A (VacA) proteins. Inhibition of IL-8 secretion was measured using enzyme-linked immunosorbent assay (ELISA) to determine the effects of sulglycotide on inflammation. Sulglycotide did not inhibit the growth of *H. pylori*, however, after six and 12 hours of infection on AGS cells, *H. pylori* adhesion was significantly inhibited by approximately 60% by various concentrations of sulglycotide. Sulglycotide decreased *H. pylori* virulence factor (CagA and VacA) translocation to AGS cells and inhibited IL-8 secretion. Sulglycotide inhibited *H. pylori* adhesion and inflammation after infection of AGS cells in vitro. These results support the use of sulglycotide to treat *H. pylori* infections.

## 1. Introduction

*Helicobacter pylori* (*H. pylori*) is a helical Gram-negative bacterium that penetrates gastric mucosa and colonizes the stomach [1]. *H. pylori* infection is associated with numerous gastric diseases [2,3], initiates acute inflammatory reactions and progresses chronic inflammatory reactions, atrophic gastric mucosal changes, intestinal metaplasia, dysplasia, and finally carcinogenesis, which is known as the inflammation-carcinoma chain [4]. Therefore, *H. pylori* is classified as a type I carcinogen [1]. *H. pylori* infection might persist for many years without inducing gastrointestinal symptoms, even though 1% to 3% of infected human hosts develop gastric cancer [5,6]. The prevalence of *H. pylori* infection exceeds half of the world’s population. In South Korea, the prevalence is 50% to 80% in 40–50-year-olds with approximately 12~49% of 16–19-year-olds infected [7], as such, gastric diseases including gastric cancer is likely to be high in South Korea.

The *H. pylori* lipopolysaccharide has a particularly significant role in virulence [8,9]; it stimulates the upregulation of pro-inflammatory cytokines, excessive generation of nitric oxide, and regression of regulatory cytokines [8,10]. In previous studies, two *H. pylori*-associated virulence factors were identified: cytotoxin-associated protein A (CagA) and vacuolating cytotoxin A (VacA). Following *H. pylori* infection, CagA infuses the gastric mucosa through type IV secretion systems and is phosphorylated, activating a kinase cascade, which results in cellular changes and inflammatory cytokine production [11,12,13]. VacA, secreted by the type V secretion system, induces cytoplasmic vacuole formation [11,14,15]. During translocation to the host, VacA interacts with host-cell mitochondria, which activates the caspase cascade and apoptosis [16,17,18,19]. Upon *H. pylori* infection recognition, major inflammatory cytokines such as interleukin (IL)-1β, IL-6, IL-8, and tumor necrosis factor alpha (TNF-α) are generated and induce inflammation. The abundance of the pro-inflammatory cytokines is associated with the level of chronic gastritis; this association diminishes in the presence of gastric atrophy or intestinal metaplasia [20]. IL-8 is a neutrophil-activating chemokine in gastric epithelial cells, making it particularly important in *H. pylori* infection [21].

Gastric mucin is an essential component in the mucosal barriers against a variety of insults. The mucosal defense system protects from mucosal injuries to maintain elasticity and permselectivity of the mucus gel and interferes with *H. pylori* colonization [22,23]. The maintenance of the mucous layer’s mucosal defense system and mucinous contents are affected by diverse gastroprotective agents including sulglycotide [24]. Sulglycotide is derived from pig duodenal mucin and is produced by extensive proteolysis. Esterification of its carbohydrate chains with sulfate groups generates a potent gastroprotective agent for the treatment of various gastric diseases [10]. Previous studies have shown that sulglycotide modulates the extent of gastric mucus inflammation via the lipopolysaccharide-mediated nuclear factor kappa B (NF-κB) signaling pathway [25] and downregulated nitric oxide synthase 2 and caspase 3 [26].

Recently, the failure rate of treatment for *H. pylori* infection is high due to antibiotic resistance, although the success of the treatment depends on several factors such as smoking or patient compliance [27,28]. In particular, clarithromycin resistance in Asia has greatly increased from 15.28% in 2009 to 32.46% in 2014 [29]. Furthermore, metronidazole resistance has already been detected at rates of more than 15% worldwide (America 44%, Africa 92%, Asia 37%, and Europe 17%), and the levofloxacin resistance rate is more than 15% in Europe (24%) [30]. Therefore, new therapeutic combinations are needed to minimize the resistance of *H. pylori*. In this regard, we investigated the inhibitory effects of sulglycotide on adhesion and inflammation following *H. pylori* infection in human gastric adenocarcinoma cells (AGS cells).

## 2. Results

A broth dilution test was performed to investigate the effect of sulglycotide on the growth of *H. pylori.* Mueller–Hinton broth containing 25, 50, 75, 100, and 200 mg/L concentrations of sulglycotide were prepared, and *H. pylori* was grown for 72 hours at each of these concentrations. Sulglycotide did not inhibit the growth of *H. pylori* at any of the tested concentrations, but growth varied depending on the concentration (Figure 1).

A urease assay was performed to determine the quantity of *H. pylori* that adhered to the surface of the AGS cells. Sulglycotide did not affect the urease activity of *H. pylori* without AGS cells. *H. pylori*-only infected the control, sulglycotide-pretreated *H. pylori*, and clarithromycin-pretreated *H. pylori* was added to the AGS cells. After six, 12, or 24 hours, *H. pylori* noninfected AGS cells lacked a urease reaction. Compared to the *H. pylori*-only infected control, bacterial adhesion to AGS cells was inhibited in a sulglycotide dose-dependent manner. Particularly, bacterial adhesion was reduced to 39, 44, or 69 percent in bacteria pretreated with 200 mg/L sulglycotide for six, 12, or 24 hours of infection, respectively (Figure 2). Sulglycotide treatment with *H. pylori* leads to highly significant adhesion inhibition (*p* < 0.001). Although the bacterial adhesion level was similar at six and 12 hours, the inhibition of adhesion after six hours of infection was more similar to the clarithromycin pretreated *H. pylori* infected controls (Figure 2A,B). It was also confirmed that the colony-forming units (CFUs) of bacteria attached to the cells significantly decreased by sulglycotide during six hours of infection (Figure 3).

To investigate the protein level of bacterial proteins in AGS cells, AGS cells were infected with *H. pylori* that had been pretreated with sulglycotide for one hour. After six hours, the cell lysates were harvested and subjected to western blotting. The CagA and VacA proteins were translocated to *H. pylori*-infected cells, and both CagA and VacA secretion decreased following sulglycotide treatment (Figure 4).

Chemokine IL-8 secretion is induced by *H. pylori* and plays a central role in gastritis pathogenesis. AGS cells were infected with *H. pylori* pretreated with sulglycotide and assessed for IL-8 protein abundance to evaluate the anti-inflammatory effects of sulglycotide on *H. pylori* infection. IL-8 expression in AGS cells infected with *H. pylori* was inhibited by sulglycotide (Figure 5).

## 3. Discussion

An inflammatory tumor microenvironment is associated with cancer progression [31,32]. *H. pylori* infection increases chronic inflammation in gastric microenvironments and upregulates the production of pro-inflammatory cytokines such as TNF-α, IL-1β, IL-6, and IL-8 through an influx of immune cells (e.g., macrophages) in the gastric mucosa [33]. These cytokines bind to specific receptors expressed on gastric epithelial cells; overexpression induces uncontrollable gastric epithelial cell proliferation, resulting in tumorigenesis and the development of gastric cancer [34,35]. Therefore, it is vital to minimize gastric mucosa damage following *H. pylori* infection to maintain mucosal homeostasis. Acquired antibiotic resistance by *H. pylori* following infection is a major concern in South Korea and worldwide. Therefore, it is necessary to identify new agents that minimize resistance of the treatment for *H. pylori* infection.

This study shows the inhibitory effects of sulglycotide on the adhesion and inflammation of *H. pylori*. Sulglycotide did not have a direct effect on *H. pylori* growth, therefore, we assumed that it would disrupt colonization of *H. pylori* in the gastric epithelium. Sulglycotide itself was not affected by urease activity of *H. pylori* in contrast to the report by Slomiany et al. [36]. Using a urease assay, we found that *H. pylori* adhesion to AGS cells was inhibited by sulglycotide in a dose-dependent manner, which was consistent with the results of counting the number of bacteria attached to the cells. Additionally, adhesion inhibition observed in cells six hours post-infection was more similar to the clarithromycin pretreated *H. pylori* infected control than the cells 12 hours after infection. Therefore, the most effective, six hours of infection treated with sulglycotide, was applied to further experiments. Collectively, these data indicate that sulglycotide has an effect on inhibiting the initial stage of the colonization process.

Inhibition of bacterial adhesion decreases the entry of bacterial proteins into host cells. *H. pylori* strains harbor multiple virulence factors such as CagA and VacA that increase inflammation of the gastric mucosa and injuries to gastric epithelial cells [37]. In *H. pylori*-infected AGS cells treated with sulglycotide, the levels of CagA and VacA proteins decreased, suggesting that sulglycotide diminishes the translocation of CagA and VacA. When *H. pylori* infects the stomach, major inflammatory cytokines are upregulated in the gastric epithelial cells, causing gastric and duodenal ulcers to develop. A previous study has reported that IL-8 is one of the early pro-inflammatory cytokines produced by gastric epithelial cells in response to *H. pylori* infection [21]. Through enzyme-linked immunosorbent assay (ELISA), we showed that sulglycotide decreases the level of IL-8, supporting the fact that sulglycotide impacts anti-inflammatory reactions to *H. pylori* infection.

In conclusion, sulglycotide inhibited bacterial adhesion and inflammation after *H. pylori* infection in AGS cells in vitro. This might mean that sulglycotide has an effect on inhibiting the initial stage of the colonization process. These results support that sulglycotide is a potential therapeutic option for the treatment of *H. pylori* infection and future in vivo studies and clinical trials are needed.

## 4. Methods

### 4.1. Materials

*H. pylori* reference strain was purchased from American Type Culture Collection (ATCC 49503, Manassas, VA, USA). Mueller–Hinton broth and Brucella agar were purchased from Becton-Dickinson (Braintree, MA, USA). Bovine serum was purchased from Gibco (Long Island, NY, USA). AGS gastric adenocarcinoma cells (ATCC CRL-1739) were purchased from ATCC. Clarithromycin was purchased from BRL Life Technologies (Grand Island, NY, USA) for adhesion assays. Protease inhibitor cocktails were obtained from Sigma-Aldrich (Saint Louis, MO, USA). Antibodies to detect CagA, VacA, and β-actin were purchased from Santa Cruz Biotechnology (Dallas, TX, USA). A human IL-8 ELISA kit was purchased from Invitrogen (Carlsbad, CA, USA). Sulglycotide (Gliptide^®^) was kindly donated by the Samil Pharmaceutical Company.

### 4.2. Bacterial and Mammalian Cell Culture

*H. pylori* was cultured on Brucella agar supplemented with 10% bovine serum and incubated at 37 °C (100% humidity) for 72 hours in microaerophilic conditions with 5% CO_2_. AGS cells were cultured in Dulbecco’s Modified Eagle’s Medium (DMEM) supplemented with 10% fetal bovine serum (FBS), 100 μg/mL streptomycin, and 100 U/mL penicillin. AGS cells were incubated at 37 °C in a humidified atmosphere with 5% CO_2_.

### 4.3. Broth Dilution Method

Bacterial colonies were collected and suspended in Mueller–Hinton broth supplemented with 10% bovine serum for cultivation. The number of *H. pylori* cells was set to McFarland 0.33 (1 × 10^8^ CFU/mL) and incubated at 37 °C for three days in a humidified atmosphere with 5% CO_2_. Bacterial growth was measured at a wavelength of 600 nm. The control group was inoculated with bacteria in medium without sulglycotide and then cultured in the same manner as the experimental group. All experiments were repeated three times.

### 4.4. Urease Activity Test

*H. pylori* was treated with the indicated concentrations of sulglycotide (25, 50, 75, 100, and 200 mg/L). After 72 hours, 100 μL of the medium was collected, centrifuged at 3000 rpm for 10 minutes, and then separated into the supernatant and cells. The supernatant was quantified by measuring the optical density at 600 nm using a NanoQuant Infinite M200. After adding 5 μL of 20% urea to the specimens respectively, the specimens were incubated at 37 °C for 10 minutes. Urease activity was determined by using an Asan Set Ammonia Kit (Asan Pharmaceutical, Seoul, Korea) to measure the ammonia levels, according to manufacturer’s instructions.

### 4.5. Adhesion Test

AGS cells were seeded in 6-well culture plates at a density of 2 × 10^5^/2 mL cells in DMEM to form a confluent monolayer, and infected with *H. pylori* at a multiplicity of infection (MOI) of 200 that had been pretreated with various doses (25, 50, 75, 100, and 200 mg/L) of sulglycotide for one hour. Two control groups were included: noninfected AGS cells and AGS cells infected with *H. pylori* pretreated with clarithromycin at a concentration of 8 μg/mL. After an indicated period of infection, the monolayer was washed twice with phosphate-buffered saline (PBS) to remove the unattached bacteria. To determine the number of bacteria attached to cells, cells were lysed by 0.1% saponin followed by incubation for five minutes at room temperature and was spread on Brucella agar supplemented with 10% bovine serum. The CFU was counted after three days of incubation. The urease test was performed by adding 2 mL of urease test solution (7 mM phosphate buffer, pH 6.8, 110 mM urea, 10 mg/L phenol red) to each sample. After 60 min, the absorbance values at 540 nm were recorded using a Nano-Quant spectrophotometer [38].

### 4.6. Western Blotting

AGS cells were washed twice with PBS and then lysed with a radioimmunoprecipitation assay (RIPA) buffer containing a protease inhibitor cocktail. The cell lysates were incubated on ice for 10 min and then centrifuged, and the supernatant was collected. Thirty μg of CagA and VacA proteins and 10 μg of UreA, UreB, and β-actin proteins were loaded on a sodium dodecyl sulfate (SDS)-polyacrylamide gel. The proteins were separated using SDS-polyacrylamide gel electrophoresis and transferred to a nitrocellulose membrane. The membrane was incubated at optimal concentrations with the primary antibody (CagA, VacA, UreA, UreB, and β-actin) at 4 °C overnight, and then incubated with the appropriate secondary antibody (anti-mouse or anti-rabbit) for two hours at room temperature. The immune-labeled proteins were visualized using enhanced chemiluminescence (ECL). β-actin was used as an internal control for mammalian cell proteins.

### 4.7. Enzyme-Linked Immunosorbent Assay (ELISA)

AGS cells were infected with *H. pylori* that had been pretreated with sulglycotide (25, 50, 75, 100, and 200 mg/L). After one hour of infection, cells were harvested to measure IL-8 cytokine abundance. AGS cells were washed twice with PBS, lysed with RIPA buffer, and centrifuged; the supernatant of each sample was collected. IL-8 concentrations were determined using a human IL-8 ELISA kit. This assay used a solid-phase sandwich enzyme immunoassay that included monoclonal antibodies raised against the respective cytokines. All procedures were performed using the manufacturer’s protocol. Cytokine concentrations of unknown samples were determined by interpolation using a standard curve based on recombinant IL-8 supplied by the manufacturer.

### 4.8. Statistical Analysis

Each experiment was repeated in triplicate and data in the bar graphs were presented as mean ± standard error of mean (SEM). All statistical analyses were performed using GraphPad Prism 6 software (GraphPad Software, San Diego, CA, USA). All data were analyzed with one-way analysis of variance (ANOVA) and *p* < 0.05 was considered to be statistically significant (* *p* < 0.05, ** *p* < 0.01, and *** *p* < 0.001).

## Figures and Tables

**Figure 1 ijerph-17-02918-f001:**
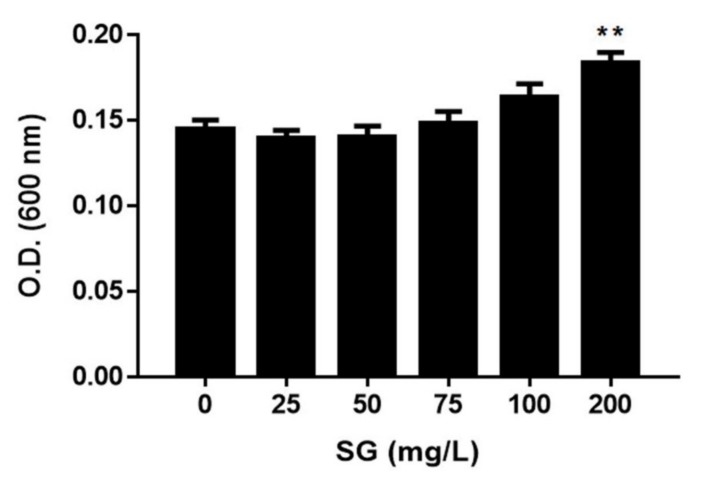
Effect of sulglycotide on the growth of *H. pylori*. *H. pylori* was grown in Mueller–Hinton broth containing indicated concentrations of sulglycotide (25, 50, 75, 100, and 200 mg/L). After 72 hours of incubation, the final optical density of the bacterial suspension was measured at a 600 nm wavelength by spectrophotometry. Results from triplicate experiments were analyzed by the Student’s *t*-test (** *p* < 0.01 and *** *p* < 0.001). O.D.; optical density, SG; sulglycotide.

**Figure 2 ijerph-17-02918-f002:**
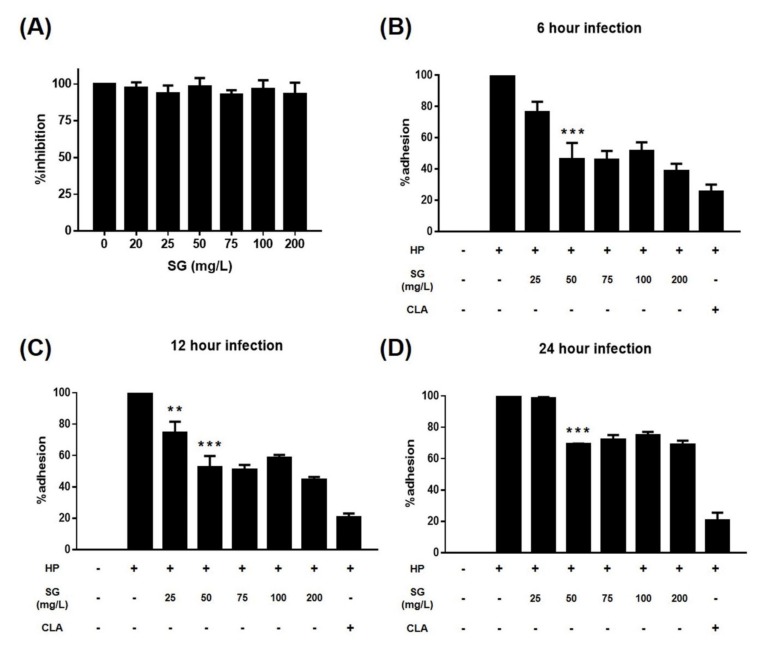
Inhibitory effect of sulglycotide on *H. pylori* adhesion to AGS cells. (**A**) AGS cells were infected with *H. pylori* (200 MOI) that had been pretreated with indicated concentrations of sulglycotide (25, 50, 75, 100, and 200 mg/L) for one hour. After (**B**) 6, (**C**) 12, or (**D**) 24 hours of infection, the urease test was performed to analyze the quantity of bacteria that adhered to the surface of the AGS cells. Results from triplicate experiments were analyzed by the Student’s *t*-test (* *p* < 0.05, ** *p* < 0.01, and *** *p* < 0.001). AGS; human gastric adenocarcinoma, HP; *H. pylori*, MOI; multiplicity of infection, SG; sulglycotide, CLA; 8 μg/mL of clarithromycin.

**Figure 3 ijerph-17-02918-f003:**
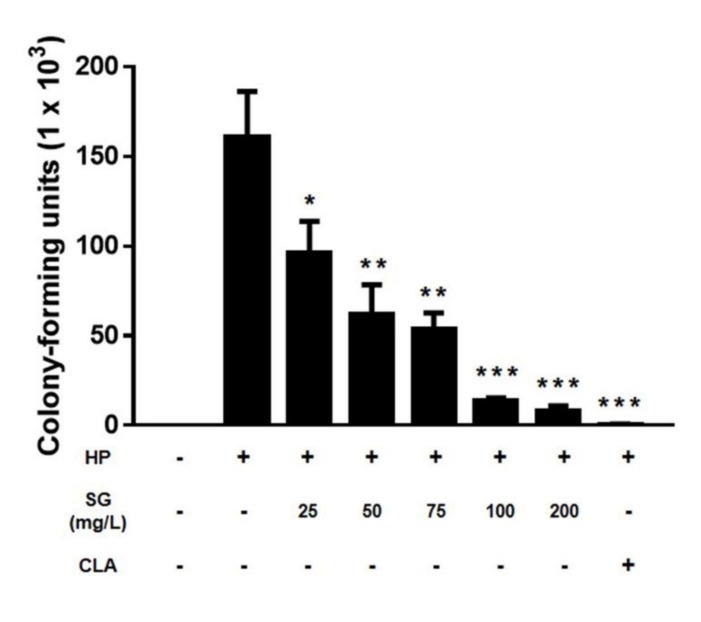
**Reduction of the number of *H. pylori* attached to AGS cells by sulglycotide.** AGS cells were infected with *H. pylori* (200 MOI) that had been pretreated with indicated concentrations of sulglycotide (25, 50, 75, 100, and 200 mg/L) for one hour. After six hours of infection, the cells were lysed with 0.1% saponin and the lysates were spread on the agar plates. After three days, the number of colonies was counted. AGS; human gastric adenocarcinoma, HP; *H. pylori*, MOI; multiplicity of infection, SG; sulglycotide, CLA; 8 μg/mL of clarithromycin. (* *p* < 0.05, ** *p* < 0.01, and *** *p* < 0.001).

**Figure 4 ijerph-17-02918-f004:**
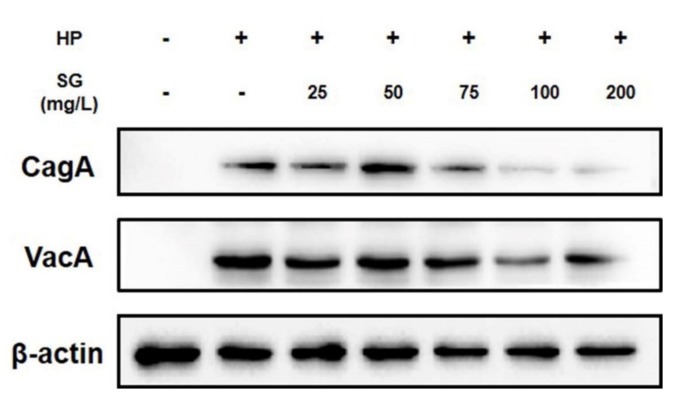
**Inhibitory effect of sulglycotide on translocation of virulence factors to AGS cells by *H. pylori*.** AGS cells were infected with *H. pylori* (200 MOI) that had been pretreated with indicated concentrations of sulglycotide (25, 50, 75, 100, and 200 mg/L) for one hour. After six hours, the cell lysates were subjected to western blotting to detect CagA and VacA proteins. β-actin was used as an internal control. AGS; human gastric adenocarcinoma, CagA; cytotoxin-associated protein A, HP; *H. pylori*, MOI; multiplicity of infection, SG; sulglycotide, VacA; vacuolating cytotoxin A.

**Figure 5 ijerph-17-02918-f005:**
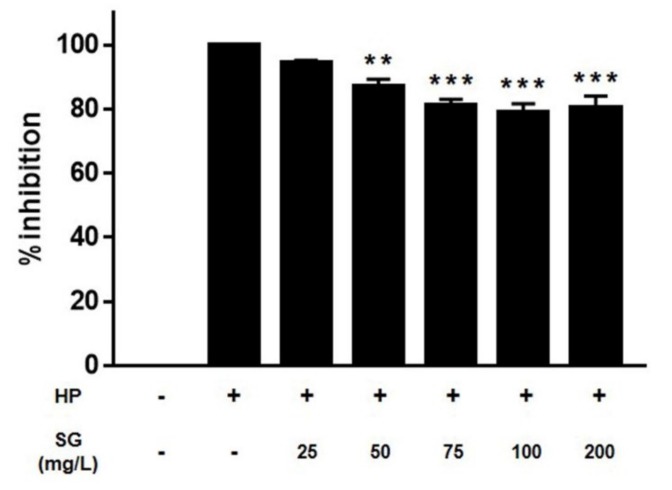
**Sulglycotide decreases IL-8 secretion in the supernatant.** AGS cells were infected with *H. pylori* (200 MOI) that had been pretreated with indicated concentrations of sulglycotide (25, 50, 75, 100, and 200 mg/L) for one hour. Culture medium was collected after six hours. ELISA was performed to determine the amount of IL-8 secretion. Results from triplicate experiments were analyzed by the Student’s *t*-test (** *p* < 0.01 and *** *p* < 0.001). AGS; human gastric adenocarcinoma, ELISA; enzyme-linked immunosorbent assay, HP; *H. pylori*, MOI; multiplicity of infection, SG; sulglycotide.

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
