# Peer review of "The Effects of Sulglycotide on the Adhesion and the Inflammation of Helicobacter Pylori"

_ijerph, 2020, doi:10.3390/ijerph17082918_

Round 1
Reviewer 1 Report
The authors answered to all questions.
This manuscript is a resubmission of an earlier submission. The following is a list of the peer review reports and author responses from that submission.
Round 1
Reviewer 1 Report
The main theme of the manuscript of Ji Yeong Yang et al. is the effect of sulglycotide on the adhesion of H. pylori to AGS cells as well as the effect of sulglycotide in inflammation when AGS cells were infected with the microorganism. The data obtained might be interesting, however, it is necessary to clarify some aspects as suggested in order to improve the manuscript. In conclusion, a major and substantial revision is suggested.
Comments:
1. Title
The title is clearly and precisely reflects the finding of the manuscript, however H. pylori should be written in italics.
2. Abstract
The abstract is clear and is in line with the context however:
- H. pylori should be written in italics (H. pylori)
- Figures should not be mentioned in the abstract
- The Authors should specify always AGS cells or “human gastric adenocarcinoma cells” instead of “gastric epithelial cells” since the reader may be misled and think the authors have used normal gastric cells.
3. Introduction
- Please add “might” persist many years without inducing..….(line 41)
- The authors should improve the introduction adding more comments regarding the problem of the increasing antimicrobial resistance particularly referred to H. pylori to emphasize the importance of studying new compounds that can be effective in inhibiting the adhesion and colonization of the microorganism.
4. Methods
Section 4.2:
Please, write “the number of pylori cells” instead “bacterial particles in the H. pylori suspension” (line 148) Please specify 0.33 McFarland how many Colony Forming Units (CFU)/mL are (line 148) In the abstract the authors wrote that the “Sulglycotide-mediated growth inhibition of pylori was evaluated using the broth dilution method”, but the method does not appear in the methods section. Please add it.Section 4.3:
The section is not clear. The authors should specify the concentrations of sulglycotide used in the experiment.Based on what did the authors choose the doses used in the experiment? Is there any bibliographical note to refer to? (line 156) The urease test used by the authors is not clear. Please explain better the method and add a reference (lines 159-162).
The Authors should perform H. pylori CFU counts to determine the effect of sulglycotide on the adesion. In particular, the CFU count should be performed both in the experiment in which the adhesion of H. pylori was evaluated, after treatment with sulglycotide, through the method of the microdilution and in the experiment in which the effect of sulglycotide was determined on H. pylori sulglycotide -treated cells co-incubated with AGS.
Section 4.5
The authors should specify at which concentration pylori had been treated with sulglycotide (line 164). Moreover the authors should specify how many independent experiments performed in duplicate or triplicate have been done. The authors should add a section regarding the Statistical Analysis
5. Results and Discussion
- Slomiany et al. 1997 demonstrated that sulglycotide is capable of reducing H. pylori urease activity at 10 micrograms/ml. In the present paper the authors demonstrated that sulglycotide inhibit the adhesion of the microorganism when in contact with AGS cells. The authors should:
Carried out the urease test also on H. pylori grown without AGS cells and comment the results obtained referring to Slomiany et al. 1997. Confirm the reduction in the adhesion with H. pylori CFU counts and microscopic images. A comparison between the controls and the treated samples (AGS cells infected with H. pylori sulglycotide-treated) is needed. Do the authors have information about the effect of sulglycotide on the expression of other cytokines, such as IL-6, TNF-α, IL-1β? As the authors explain the fact that the sulglycotide does not inhibit the adhesion of H. pylori in vitro but inhibit it when the bacterium is in contact with AGS cells? The authors should provide an overview of the data present in the literature concerning the efficacy of sulglycotide in the prevention of colonization and in the virulence of pylori. Moreover, the authors should emphasize the novelty of the work, adding experiments that differentiate it from the works already present in the literature as well as discussing the papers already published .
Biochem Mol Biol Int. 1993 Apr;29(5):965-71.
Czajkowski A,Piotrowski J,Yotsumoto F,Slomiany A,Slomiany BL.
Inhibition of Helicobacter pylori colonization by an antiulcer agent,sulglycotide.
Biochem Mol Biol Int.1995 Mar;35(3):467-72.
Piotrowski J, Murty VL, Slomiany A,Slomiany BL.
Susceptibility of Helicobacter pylori to antimicrobial agents: effect of sulglycotide.
6. Figure legends
Please, the authors should move the explanation of the method in the section Materials and Methods.
7. References
Please, write the names of Bacterial species and bacterial genes in italics -Please check the references accurately, according to the journal guidelinesAuthor Response
Please see the attachment.

Reviewer 2 Report
The present work is easy to read and understood although the authors should consider some corrections.
In abstract at results section the references to figures have to be deleted. They authors should rewritten the conclusion because it is extremely direct and they have to consider that their data are very preliminary.
In introduction the prevalence of H.pylori infections must be reviewed, because the range of years is wrong according the reference 7.
In order to understand correctly the results, the authors should talk directly of adhesion porcentage in stead of inhibition of adhesion.
The authors should discussion why there is more CagA and VacA in AGS cells treated with 50 mg/l of SG, because this result is very confuse.
With respect to material and methods section the authors should explain the H.pylori treated with sulglycotide. Also, they must annote how many protein total grams are used to detected the virulence factors by western blotting. A section of stadistical analysis must be included, moreover they must analyze the results with one-ANOVA test, because they must analyze whether there is any effects between the differents doses.
The discussion should be reviewed considering the above comments and the conclusion has to be softened
Round 2
Reviewer 1 Report
Although the authors answered most of the questions. The fact remains that the authors did not perform the experiments required for confirming the effect of Sulglycotide on the adhesion of H. pylori. Slomiany et al., 1997 demonstrated that sulglycotide at 10 micrograms/ml induces a decrease in urease activity corresponding to 70.2 % of inhibition. To demonstrate that the Sulglycotide inhibits the adhesion of the microorganism the urease test is not suitable or however must be supported by other methods as the CFU counts and/or immunofluorescence assay using specific anti-H. pylori antibodies . The CFU counts or the immunofluorescence assay using specific anti-H. pylori antibodies is mandatory for the acceptance of the manuscript.
Reviewer 2 Report
The authors have appropiately answered to all the suggestions, So this present version is acceptable to publish in this journal.